# Cellular Strategies for Separating GvHD from GvL in Haploidentical Transplantation

**DOI:** 10.3390/cells13020134

**Published:** 2024-01-11

**Authors:** Mauro Di Ianni, Carmine Liberatore, Nicole Santoro, Paola Ranalli, Francesco Guardalupi, Giulia Corradi, Ida Villanova, Barbara Di Francesco, Stefano Lattanzio, Cecilia Passeri, Paola Lanuti, Patrizia Accorsi

**Affiliations:** 1Hematology Unit, Pescara Hospital, 65124 Pescara, Italy; carmine.liberatore@asl.pe.it (C.L.); nicole.santoro@asl.pe.it (N.S.); paola.ranalli@unich.it (P.R.); 2Department of Medicine and Aging Sciences, University of Chieti-Pescara, 66100 Chieti, Italy; francescoguardalupi@gmail.com (F.G.); giulia.corradi@unich.it (G.C.); stefano.lattanzio@unich.it (S.L.); paola.lanuti@unich.it (P.L.); 3Center for Advanced Studies and Technology (CAST), University of Chieti-Pescara, 66100 Chieti, Italy; 4Blood Bank Unit, Pescara Hospital, 65124 Pescara, Italy; ida.villanova@asl.pe.it (I.V.); barbara.difrancesco@asl.pe.it (B.D.F.); cecilia.passeri@asl.pe.it (C.P.); patrizia.accorsi@asl.pe.it (P.A.)

**Keywords:** GvHD, GvL, TCR alpha-beta, CD45RA, Tregs, NK, suicide gene, immunotherapies

## Abstract

GvHD still remains, despite the continuous improvement of transplantation platforms, a fearful complication of transplantation from allogeneic donors. Being able to separate GvHD from GvL represents the greatest challenge in the allogeneic transplant setting. This may be possible through continuous improvement of cell therapy techniques. In this review, current cell therapies are taken into consideration, which are based on the use of TCR alpha/beta depletion, CD45RA depletion, T regulatory cell enrichment, NK-cell-based immunotherapies, and suicide gene therapies in order to prevent GvHD and maximally amplify the GvL effect in the setting of haploidentical transplantation.

## 1. Introduction

Graft-versus-host disease (GvHD) is one of the most serious complications of allogeneic stem cell transplantation [1] and it is tied up to the action of the T lymphocytes present in the inoculum of stem cells which attack the host’s tissues, giving an extremely polymorphic and potentially fatal clinical complication [2]. On the other hand, the T lymphocytes themselves are mainly responsible for the antileukemic effect (i.e., graft versus leukemia, GvL) associated with the transplant procedure [3]. In the setting of allogeneic transplantation from a haploidentical donor, the first platform for GvHD prevention was based on profound nonselective T depletion, where all T cells were removed from the hematopoietic stem cell (HSC) inoculum and which, however, compromised the immunological reconstitution and the same GvL effect [4,5]. Since then, different approaches have been proposed, but the separation between GvHD and GvL still remains an open challenge [6,7]. In this review, we will summarize the more recent cellular strategies aimed at separating GvHD from GvL by using selective T cell targeting for patients undergoing stem cell transplantation for hematological malignancies.

## 2. Removal of αβ T Cells

The T cell receptor (TCR) is made up of two different chains. The majority (95%) of circulating peripheral blood T lymphocytes express αβ chains, while the remainder has γδ chains. In the setting of allogeneic HSCT, experimental models have clearly demonstrated how αβ T lymphocytes are implicated in the generation of GvHD. Conversely, γδ T lymphocytes that recognize antigens in an MHC-independent manner are involved in facilitating engraftment, driving immune reconstitution, reducing the risk of opportunistic infections, and potentially exerting antileukemic effects [8]. Clinical studies have clearly demonstrated that a higher number of γδ lymphocytes in the graft correlates with better disease-free survival [9,10]. Hence the rationale for removal of αβ T cells from the graft with the aim of limiting GvHD and preserving at the same time immunological reconstitution and GvL, thus providing a valid alternative for those patients requiring an urgent allogeneic HSCT but lacking full-matched donors. A large-scale and efficient method was developed for the selective ex vivo depletion of alloreactive αβ T cells and CD19^+^ B cells from mobilized peripheral blood stem cells and generation of an allogeneic graft enriched for CD34^+^ stem cells and γδ T lymphocytes [11,12,13]. Moreover, differently from the positive selection of CD34^+^ cells, procedures of αβ T-cell depletion allowed the sparing of donor-derived NK cells, a subset of cells fundamental in both GvL and control of opportunistic infections [14,15,16,17]. The first applications of αβ T-cell depletion involved pediatric haploidentical HSCT for both malignant and nonmalignant diseases. After removal of αβ T cells and CD19^+^ B cells, haploidentical HSCT proved feasible and effective in children with life-threatening nonmalignant disorders. A total of 4 out of 23 patients experienced graft failure that was successfully resolved with re-transplantation, while the remaining children had a rapid hematopoietic recovery. Notably, only three children suffered from skin-limited and grade I–II acute GvHD, whereas none of them had severe acute nor chronic GvHD. After a median follow up of 18 months, cumulative incidence of transplantation-related mortality was 9.3% and 2-year disease-free survival was 91.1%, comparing favorably with outcomes of HSCT from HLA-matched donors and cord blood unit [18,19]. Reduced incidence of graft failure (14%) was observed in patients with hemoglobinopathies, a setting where engraftment historically represented a relevant obstacle to HSCT [20]. Comparable results between αβ T cells and CD19^+^ B cells depleted MUD and mismatched related donors were also confirmed in a prospective trial enrolling pediatric patients with primary immunodeficiencies [21]. In children with either high-risk or relapsed acute leukemia lacking suitable full-matched donors, an αβ T-cells- and B-cells-depleted haploidentical HSCT was given following myeloablative conditioning regimen. Anti-T-lymphocyte globulin was given for preventing graft rejection and GvHD, whereas no further GvHD prophylaxis was given post infusion. Among 80 enrolled patients, primary graft failure occurred in only 2 children. At day 100, skin-only and grade I/II acute GvHD occurred in 24 patients (30%), whereas no case of severe and gut/liver acute GvHD was reported. Among patients surviving >100 days after HSCT, CI of chronic GvHD was 5% and all cases were limited in severity. After a median follow up of 46 months, CI of relapse and non-relapse mortality was 24% and 5%, respectively. The 5-year OS and GvDH-free relapse-free survival (GRFS) was 72% and 71%, respectively, comparing favorably with outcomes of HSCT from both MRD and MUD [22]. Similar results following a treosulfan-based conditioning regimen were obtained in patients receiving αβ T-cells- and B-cells-depleted unrelated as well as haploidentical HSCT [23]. Among pediatric patients with both malignant and nonmalignant disease, αβ T cell and CD19^+^ B cell depletion granted efficient control of opportunistic infections. Although detection of CMV and EBV viremia occurred in 51% and 33% of patients, respectively, the incidence of CMV-associated disease was 6%, whereas EBV-related disease involved 0.5% of patients [24]. Moreover, compared to CD34^+^ selected grafts, those patients who received αβ T-cells- and CD19^+^ B-cells-depleted haploidentical HSCT also showed more rapid immune recovery in terms of CD3^+^, CD19^+^, and CD56^+^ counts, with γδ T cells representing the prevalent T-cell subset in the early post-transplant period [8,25]. Notably, a robust recovery of γδ T cells at early timepoints appeared to correlate with decreased risk of CMV infection and leukemia relapse [26]. Following results in children, αβ T cell and CD19^+^ B cell depletion was then employed in adults. Following a conditioning regimen including ATG, treosulfan, fludarabine, and thiotepa without further GvHD prophylaxis post-transplantation, 59 adult patients with acute leukemia underwent haploidentical HSCT. Only three patients (95%) experienced graft failure, whereas rapid full-donor hematological engraftment as well as sustained immune reconstitution were observed among the remaining patients. Grade II/IV acute GvHD was limited to two cases, while two patients developed chronic GvHD. Disease relapse and non-relapse mortality remained the main causes of treatment failure [27]. Similar results were reported in a Turkish experience [28]. More recently, αβ T cell depletion has been tested even in the setting of HSCT from matched related (MRD) and matched unrelated donors (MUD). In a phase 1/2 prospective trial, 35 adult patients with hematological malignancies underwent peripheral-blood-derived αβ T-cells-depleted allogeneic HSCT. At day 100, cumulative incidence (CI) of grade II–IV and grade III–IV acute GvHD was 26% and 14%, respectively. Rapid immunological reconstitution by NK and γδ T cells was observed and the majority of patients could also receive DLI. At 2 years, CI of moderate and severe chronic GvHD was 17% and 0%, respectively, while incidence of relapse and non-relapse mortality was 29% and 32%, respectively [29]. Procedures of αβ T cell depletion have been applied for the manipulation of DLI and stem-cell booster for the treatment of poor graft function, mixed chimerism, and opportunistic infections after HSCT with limited GvHD and promising results [30]. The clinical trials in the haploidentical transplantation are summarized in Table 1.

### 2.1. Removal of Naive T Cells (CD45RA T-Cell Subset)

CD45RA, an isoform of the well-known common leukocyte antigen CD45, identifies human naïve T (T_N_), which are a subtype of T cells that have yet to encounter their antigen, while T cells that previously responded to their antigen, called memory T (T_M_) cells, became CD54RA negative. In preclinical models, it has been observed that T_N_ are responsible for a more severe GvHD than T_M_, which retain more antipathogen immunity with a graft versus leukemia (GvL) activity [31,32,33,34]. These findings supported the hypothesis that eliminating T_N_ cells (CD45RA^+^) from the graft could be a potential weapon for dissecting GvHD from GvL and enhance immune reconstitution. Teschner et al. [35] firstly described the depletion of CD45RA^+^ cells from leukapheresis product of six donors using immunomagnetic beads. Post CD45RA^+^ depletion, the targeted T-cell content was 1 × 10^7^ cells/kg in the graft. This technique allowed a T_N_ depletion of median of 4 log. Based on these data, depleted CD45RA^+^ cells were investigated in the clinical setting as progenitor cell grafts and in a post-transplant setting as donor lymphocyte infusions (DLIs) to enhance immune reconstitution. Different groups reported outcomes of CD45RA^+^-depleted grafts in matched related, unrelated, and haploidentical HCT. In 2015, Bleakley et al. published the results of the first pivotal single-arm phase II clinical trial (NCT00914940) [36] including 35 adults with high-risk acute leukemia transplanted from a matched sibling donor. Conditioning regimen was myeloablative and GvHD prophylaxis was based on tacrolimus alone. CI of II–IV aGvHD was high, resulting 66%, but no steroid-refractory aGvHD was observed; only 9% of patients developed cGvHD. Two-year OS was 78% and two-year DFS was 70%. The 2-year probability of relapse was 21%. Immune reconstitution was rapid and sustained, resulting in 2 y NRM of 9%. EBV reactivation and post-HCT lymphoproliferative disease were not observed. More recently, the same group [37] reported the outcomes of 138 (adult and pediatric) patients with acute leukemia and myelodisplastic syndrome treated on three different prospective phase II single-arm trials (NCT00914940, NCT01858740, and NCT02220985) receiving T_N_-depleted peripheral graft from HLA matched related or unrelated donors. Conditioning was of high intensity for 100 patients and of intermediate intensity for 38 patients with age > 50 or comorbidities. GvHD prophylaxis was based on tacrolimus alone for patients that received matched related donor (n = 41) and on tacrolimus plus methotrexate (n = 59) or tacrolimus plus mycophenolate mofetil (n = 38) in matched unrelated donors receiving high and intermediate conditioning intensity, respectively. CI of grade II aGvHD was 71% and mostly was stage I upper gastrointestinal and steroid-responsive; CI of III–IV aGvHD was 4%. Three-year CI of cGvHD occurred in only 7% of patients but was mostly mild and steroid-responsive. No differences in acute and cGvHD were found according to the donor type. Three-year OS was 77%, cGvHD-free and relapse-free survival (GRFS) was 68%, CI of relapse was 28%, and NRM was 8%.

Overall, these results showed a low incidence in severe acute and cGvHD, without apparent risk of relapse and NRM. One possible explanation of these clinical results is that T_N_ includes a greater frequency of minor histocompatibility (H) antigen-reactive T cells, while the T_M_ remaining after the depletion of CD45RA^+^ cells have a limited TCR repertoire and could potentially recognize minor H antigens to a lower extent [38]. This might generate a sufficient alloresponse to induce limited aGvHD and some GvL but insufficient to initiate or sustain cGvHD. This hypothesis could support the fact that T_M_ favor an alloresponse which leads to limited aGvHD and is not sufficient to generate and or sustain cGvHD [39]. In haploidentical HCT, the adoptive transfer of a diverse memory T cell population from the CD45RA^+^-depleted grafts has been reported by some investigators [40,41] mostly focused on the pediatric population. Naik S et al. [42] reported results of a prospective clinical trial using CD45RA^+^-depleted haplo transplant followed by donor NK cell addback in 72 pediatric patients with hematological malignancies. All patients received submyeloablative conditioning and GvHD prophylaxis consisted of a short course of Mycophenolate mofetile and/or sirolimus. Patients received CD34^+^ selected graft at day 0 and a second progenitor graft depleted of CD45RA^+^ cells; NK cells were infused at day +6. CI of overall aGvHD was 36.1% and cGvHD was 20.8%. Three-year CI of relapse and NRM were 26.5% and 11.5%, despite the majority of patients being transplanted in a relapsed refractory setting. Sisinni et al. [43] reported outcomes of 25 pediatric patients with acute leukemia who received CD45RA^+^-depleted T cell grafts after submyeloablative conditioning. GvHD prophylaxis was based on a short course of cyclosporine and CI of II–IV aGvHD was 39% and, at 30 months, CI of cGvHD was 22%. Immune reconstitution was rapid but there was an unexpectedly high rate of HHV6 encephalitis (34% of patients) at a median 35 days after transplant. Some in vitro experiments showed that NK could eliminate HHV6 CD4^+^ T cells [44], so the same group [45] reported outcomes of 18 patients, incorporating NK cell infusion within 10 days after CD45RA^+^-depleted transplant. A total of 8 of 18 patients had HHV6 reactivation but none of the patients developed HHV6 encephalitis. Despite CD45RA^+^-depleted T-cell grafts providing a feasible transplant platform with reliable cell processing using widely available commercial technology, some points need to be elucidated especially in the haploidentical setting, such as the optimal T-cell dose and the GvHD prophylaxis. To date, there are two randomized clinical trials ongoing comparing outcomes of naive T-cell-depleted HCT to T replete transplant platform (NCT03970096 NCT03779854) both in the adult and in pediatric setting. Results will determine whether this approach could really improve the risk of acute and chronic GvHD and survival outcomes compared to the standard HCT platforms. Furthermore, CD45RA^+^ T cell depletion was also reported in a small series of nonmalignant disorders [45,46] but further data are needed in this setting. The clinical trials in the haploidentical setting are summarized in Table 2.

Importantly, CD45RA^+^-depleted T cells have also been used in the post-transplant setting as modified donor lymphocyte infusions (DLIs) with the main objective of reducing relapse incidence and enhancing immune reconstitution while preventing GvHD. The use of DLI to treat relapse was investigated by Muffly et al. [48]. They evaluated the feasibility and safety of infusing freshly isolated and purified donor-derived phenotypic CD8^+^ T_M_ cells into 15 adults with disease relapse after allo-HCT. DLI were given at escalated doses (from 1 × 10^6^/kg to 10 × 10^6^/kg) and the majority received chemotherapy before infusion. aGvHD grade II occurred in one patient. In total, 67% achieved response for at least 3 months after infusion. Median LFS was 4.9 months and OS was 19 months. Other groups focused attention mainly on the role of DLI in enhancing immune reconstitution and preventing GvHD. Dunaikina et al. [49] evaluated the safety and efficacy of CD45RA^+^-depleted prophylactic DLI given early after haplo-HCT with αβ T-cell depletion in pediatric patients with acute hematological malignancies. From a cohort of 149 children, 76 patients were randomized to receive scheduled DLI and 73 received standard care. The median number of DLI was 4 and the dose was escalated from 25 × 10^3^/kg up to 50 × 10^3^/kg. The CI of grade II–IV aGvHD, the incidence of CMV viremia, and survival outcomes were similar in the two groups. The use of DLI was associated with improved recovery of CMV T-cell responses in a sub-cohort of CMV IgG seropositive recipients. In the same year, Naik et al. [47] reported an interim analysis of a prospective clinical trial (NCT03849651) utilizing escalating doses of CD45RA-depeleted T cells as addback following TCRαβ/CD19-depleted haplo-HCT to improve immune recovery in 30 children with acute leukemia. Patients with acute lymphoblastic leukemia (ALL) also received prophylactic Blinatumomab following infusion of CD45RA-depeleted T cells to overcome the risk of immune escape secondary to HLA loss and relapse. Two weeks after engraftment, patients received CD45RA^+^-depleted cells in three escalating doses (starting dose 1 × 10^5^/kg increasing by 1 log for each infusion). At 1 month post infusion, authors described significant expansion of virus-specific T cells and, at 6 months, TCR repertoire was broad and comparable to that of the donor. The cumulative incidence of aGvHD and grade III–IV aGvHD for the entire cohort was 26.7% and 13.3%, respectively; there was no chronic GvHD. In an adult setting, Castagna et al. evaluated the role of CD45RA^+^-depleted DLI after haplo-HCT with post-transplant Cyclophosphamide for patients with hematological malignancies [50]. DLI was delivered in three escalating doses; the median first dose was given at 55 days post-transplant. A total of 16 of 19 patients received all the three planned infusions (starting dose was 5 × 10^5^/kg up to 5 × 10^6^/kg). Only one patient had development of grade II acute GvHD and two patients had moderate chronic GvHD. The 100-day CI of viral infection was reduced (53% vs. 32%) from previously published data of the same group. Maung et al. [51] assessed the safety of prophylactic CD45RA^+^-depleted DLI after reduced-intensity conditioning transplant from T-repleted matched related or unrelated donors in 16 patients with hematological malignancies. The first dose was given at a median of 112 days (starting dose was 1 × 10^5^/kg, increasing 1 log each administration up to 1 × 10^7^). No dose-limiting grade III–IV aGvHD was observed, suggesting that prophylactic modified DLI is safe and not associated with increased risk of acute and chronic GvHD. Taken together, these findings are promising but future strategies could include more harmonized procedures and randomized clinical trials to determine whether prophylactic CD45RA^+^-depleted DLIs can improve immune recovery and reduce infectious complications without increasing the risk of GvHD. The clinical trials after CD45RA-depleted DLI in haploidentical setting are summarized in Table 3.

### 2.2. Tregs Selection

Regulatory T cells (Tregs) play a critical role in regulating adaptive immunity and maintaining tolerance [52]. Tregs exhibit a CD4^+^/CD25^+^ phenotype accounting for 5–10% of the circulating T cells [53] and up to 10% of peripheral blood CD4^+^ T cells express the CD25 antigen [54]. However, only 1–2% express high levels of CD25 (CD25^hi^) and have suppressor activity [55]. Tregs can be easily separated from leukapheresis of the donor by immunomagnetic separation (CD19 w/o CD8 depletion followed by CD25 enrichment) resulting in a population ranging from 200 to 400 million cells with the CD4^+^/CD25^+^ phenotype. These cells contain a variable amount of CD127^+^ cells (5–20%) and are strongly enriched in FoxP3 (about 90%) [56,57,58,59]. Another approach used in the setting of cord blood transplantation is based on obtaining Tregs from umbilical cord blood (UCB) by CD25 enrichment with a final purity of CD4/CD25 ≥ 60% [60]. In addition to immunomagnetic selection, some groups have also developed selection by GMP-grade cell sorter [61,62], with, however, clinical applications different than haploidentical transplantation. The main transplantation platform based on the use of Tregs cells in a haploidentical setting represents the evolution of the T-depleted haploidentical transplantation [5], in which, after a myeloablative conditioning regimen, in addition to the megadose of CD34^+^ stem cells, 1 million CD3^+^ cells/kg are infused under the protection of 2 million/kg Tregs [63,64,65] in the absence of post-transplant immunosuppression. Furthermore, a characteristic of this procedure is that the Tregs are infused 4 days before the conventional T cells (Tcons), on the basis of the experimental model [66] in which it is clearly demonstrated that the prevention of GvHD correlates with the early administration of Tregs. With such a strategy, the probability of moderate/severe cGvHD/relapse-free survival was 75% [65]. The mechanisms of action of Tregs in terms of immunosuppression and, therefore, inhibition of GvHD are multiple and include cytokine production, such as interleukin (IL)-10 [67], IL-35 [68], and TGF-beta [69]; direct killing of Tcons by perforin/granzyme mechanism [67]; IL-2 competition that causes IL-2 starvation on Tcons [67]; Tregs/DC interaction with downregulation of CD80/CD86 on DCs via CTLA-4 [70], interference with DC maturation through the LAG-3 molecule highly expressed on Tregs [71], and Treg-mediated enhanced expression of indoleamine 2,3-dioxygenase (IDO) by DCs, which accelerates tryptophan disruption fundamental for the survival of Tcons [72]; and adenosine triphosphate (ATP) cleavage by CD39/73 expressed on Tregs which transform ATP to adenosine, an anti-inflammatory factor [73] (Figure 1). However, how Tregs inhibit GvHD but not GvL has not yet been fully clarified. Mouse models have demonstrated that Tregs are able to inhibit the onset of GvHD but do not hinder the GvL effect of simultaneously co-infused T lymphocytes [64,74]. The explanation given was that Tregs inhibited the proliferation of T lymphocytes but not their activation, thus resulting in them being capable of lysing the leukemic target [74]. This observation has recently been supported by TCR receptor and transcriptome analyses, which confirmed that Tregs do not alter the activation of Tcons, thus guaranteeing the GvL effect and revealed as potential GvHD-modulating molecules IL-10 and IL35 [75]. Interestingly, both molecules are found significantly increased in GMP expanded Tregs [76]. Tregs cells are also implicated in the mechanisms of inhibition of NOTCH1, which represents a key regulator of alloreactivity [77,78] and, in this regard, it has been demonstrated that NOTCH1 is downregulated in Tcons in the presence of Tregs through a CD39-dependent mechanism in both in vitro and in vivo models [79]. Interestingly, NOTCH1 inhibition, on the one hand, blocks GvHD [80] and, on the other hand, does not alter the T-cell-mediated GvL effect [81]. STAT3 deficiency and PD1 signaling are both important for the prevention of GvHD in target organs, while, where this axis functions less, i.e., in lympho-hematopoietic tissues, T cell proliferation is maintained with a consequent powerful GvL effect [82]. The analysis of the interaction between the STAT3/PD1 axis and Tregs has yielded conflicting results and further studies are needed to establish the exact role of Tregs in the above tight regulation [82,83,84,85]. The presence of an environment more prone to inflammatory activity in the bone marrow and instead of suppression of proliferation in the target organs of GvHD is confirmed in a recent study, which demonstrates the presence of a population of the less suppressive CD161^+^ Tregs electively localized in the bone marrow of patients who have undergone a transplant with regulatory T cells [86]. Figure 1 illustrates the main mechanisms involved in controlling GvHD while sparing the GvL effect. Currently, the most used protocols are based on the use of Tregs cells selected immunomagnetically but not expanded in vitro. However, the number of Tregs that can be collected from a donor is relatively low (1 million/kg). One possibility to increase the number of infused Tregs is to use ex vivo expanded Tregs with the advantage of having large numbers of cGMP-grade Tregs [60,62,76,87,88,89]. The major obstacle is represented by the requirement of GMP manufacture, which is expensive, not always available, and requires expert, dedicated laboratory staff [90].

The success of CAR T-cell therapy in hematological cancers has sparked interest in redirecting the specificity of regulatory T cells (Tregs) towards antigens responsible for autoimmunity and transplant rejection [91,92]. In recent studies, CAR Tregs were specifically designed to address alloimmunity, focusing on the human leukocyte antigen A2 (HLA-A2) present in transplanted tissues but absent in recipients. The primary objective was to reorient the Tregs to enhance tolerance for transplanted grafts and reduce the incidence of GvHD [93,94,95,96]. Notably, anti-HLA-A2 CAR Tregs demonstrated superior performance compared to polyclonal Tregs. They were more effective in suppressing xenogeneic GvHD and significantly reducing the rejection of skin allografts [93,94,95,96]. In an early phase I clinical trial (NCT05993611) for the treatment of cGvHD, a different antigen is under investigation to redirect CAR Tregs [97]. This antigen is known as CD6 and is found primarily on the patient’s T lymphocytes. CD6 binds to activated leukocyte cell adhesion molecule (ALCAM) expressed on antigen-presenting cells (APCs) [97]. CD6 is crucial in the activation, growth, differentiation, and movement of T lymphocytes. Another antigen that has been recently examined in preclinical trials for the redirection of CAR Tregs is the CD19 antigen found on B cells [98]. The evaluation of the best CAR design for Tregs is still ongoing since different costimulatory domains could have an impact on the phenotype, function, and cytokine secretion of Tregs [99,100]. According to a study conducted by Boroughs and colleagues, using a 4-1BB-based CAR in Tregs had a detrimental impact on their ability to carry out their regulatory functions [99]. Another study by Dawson et al. conducted an extensive investigation into how various co-stimulatory domains influence the function of an anti-HLA-A2 CAR in an allotransplantation model [100,101]. Their data revealed that the CAR encoding CD28 was more effective both in vitro and in vivo concerning proliferation, suppression, and the delay of GvHD symptoms, while the presence of 4-1BB-CAR had a negative impact on Treg function and stability [100]. In contrast to Dawson and colleagues’ findings, Koristka et al. employed a modular CAR technology known as UniCAR and demonstrated that CD28-based CARs might exhibit off-target effects and enhanced cytolytic activity when compared to CARs based on 4-1BB [102].

### 2.3. NK

The therapeutic potential of donor NK cells was studied in both haploidentical transplantation and also in the nontransplantation setting. Three main sources for allogenic NK cells are available actually, each presenting advantages and disadvantages: donor peripheral blood, cord blood, and progenitor cells (HSPC) or induced pluripotent stem cells (IPSc) [103,104]. Several trials showed feasibility and safety of infusing high doses of NK cells after haploidentical HSCT with relevant benefit for hematologic malignancies. In 2014, Choi et al. [105] published the results of the first trial of donor NK cells administered at high dose after haploidentical allotransplant; this latter performed as salvage treatment in 41 patients with previous diagnosis of active/refractory hematological malignancies, mostly AML. In this study donor-derived NK cells from a mobilized leukapheresis were infused at a median dose of 2.0 × 10^8^/kg. aGvHD was reported in 22% of cases at a median of 8 months after transplant, cGvHD occurred in 24% of cases at a median of 3.3 months after HCT. As compared to patients who underwent HLA-haploidentical HCT, investigated patients obtained a significant reduction in leukemia progression (74% to 46%), providing evidence of enhanced antileukemia effect of donor NK cells, possibly with direct action on leukemia cells or action as enhancers of a T-cell-mediated antileukemia effect [106]. Notably, more NK cells expressing activating receptors were detectable early after infusion in the peripheral blood of patients who received NK infusion [106]. NK cells require homeostatic cytokine support after transfer. Intermediate or low-dose IL-2 for 2 weeks after transfer has been tested in multiple trials and they are generally considered safe, with evidence of in vivo function persistence [103]. In a first trial after infusion of donor mature NK cells followed by administration of IL-2 daily for 14 days, a successful donor NK-cell expansion was observed for patients previously treated with cyclophosphamide and fludarabine [107,108], thus allowing the achievement of complete remission in 30% of poor prognosis AML patients. Furthermore, in order to increase the antileukemic action of transplant without worsening the risk of GvHD, one can decide to infuse NK cells from an HLA haploidentical donor, chosen for its alloreactivity, and distinct from a separate HLA identical donor, chosen for allotransplantation. In this study, NK cells obtained from PBMCs of an HLA haploidentical related donor, after an overnight incubation with IL-2 were infused at escalating doses (dose ranging from 0.02 to 8.32 × 10^6^/kg) in 21 high-risk myeloid malignancies. After infusion, subcutaneous IL-2 was administered daily for 5 days. Overall, 100% engraftment and a rate of 10% of ≥grade 3 aGvHD was observed. According to this study, relapse-free, overall, and GvHD/relapse-free survival were 102, 233, and 89 days, respectively [109]. IL-15R agonists may represent a possible alternative to IL-2 [110,111] in spite of the risk of an induced cytokine-exhausted state [112]. The use of membrane-bound interleukin-21-expressing cells led to an impressive 35,000-fold expansion of natural killer (NK) cells within 21 days [113]. In a clinical study involving 13 individuals with high-risk myeloid malignancies, increasing doses of donor NK cells, expanded using these interleukin-21-expressing cells, were administered before and after haploidentical HCT (on days −2, +7, and +28). Doses ranged from 1 × 10^5^/kg to 1 × 10^8^/kg per dose, escalating up to 3 × 10^8^/kg, followed by post-transplant cyclophosphamide to prevent severe GvHD. Approximately half of the cases experienced mild to moderate acute GvHD (Grade 1–2), while severe aGvHD (Grade III–IV) or cGvHD were notably absent. Relapse rate and overall mortality was not different than in conventional transplants, without NK cell infusion [113]. The only case of relapse happened at day +120 post-transplant and it was observed in a patient who received the lowest investigated dose (1 × 10^5^/kg per dose). One-year OS and DFS are 92% and 85%, respectively [114]. Memory NK cells are being used as an adjunct to haploidentical transplants for patients with advanced AML [115]. Human memory-like (CIML) NK cells, previously preactivated with a combination of IL-12, IL-15, and IL-18 [116,117], could represent another population able to induce a powerful GvL effect [118] in the absence of GvHD. A total of 15 adult patients with high-risk relapsed/refractory AML were infused with a range of 0.5 × 10^6^ to 10 × 10^6^ cells/kg after RIC conditioning. The infusion was performed on day +7 after transplant and supported by IL-15 administered subcutaneously at 10 μg/kg starting on day +7 and over 3 weeks. Tacrolimus and mycophenolate mofetil were administered as GvHD prophylaxis. Patients showed a good tolerance profile to NK cell transfer, only limited CRS happened, and only transient reactions in the site of injection were observed. Acute GvHD occurred in 10 patients (grade 1:4, grade 2:6), comparable to expected rates with RIC haplo-HCT, as well as graft failure. Chronic GvHD occurred in nearly 20% of patients. The clearance of high-risk mutations, including TP53 variants, was obtained in 87% of patients. A total of 80% of patients were alive at day +100; four patients were still in CR at that time. One-year OS was 29% [110]. Taking into consideration their rapid expansion and long-term persistence, cytokine-induced memory-like (CIML) NK cells could represent a plausible platform for the treatment of post-transplant relapse of myeloid disease. In a Phase I trial, the recurrence of myeloid malignancies (AML, MDS, MDS/MPN, or blastic plasmacytoid dendritic cell neoplasm (BPDC)) post-haploidentical HCT was addressed with lymphodepleting chemotherapy followed by the infusion of donor-derived CIML NK cells at a dosage ranging from 5 to 10 × 10^6^ cells/kg, along with IL-2 administration. This approach led to a rapid and sustained in vivo expansion of NK cells. The CIML NK cells were generated from non-mobilized apheresis products using a two-step process involving CD3^+^ depletion, followed by CD56^+^ selection. Among the initial six enrolled patients, by day +28, a favorable disease response was observed in four out of six individuals, with three out of six patients demonstrating a complete response (CR). Notably, neither aGvHD nor cGvHD was evidenced in any patient [119]. The clinical trials in a haploidentical setting are summarized in Table 4.

### 2.4. Suicide Gene Therapy

The engineering of polyclonal donor T cells with the insertion of suicide genes capable of limiting their expansion and activity is a strategy that has been adopted in the setting of allogeneic HSCT to separate GvL and GvHD. The major experience was carried out by the San Raffaele group in Milan with the use of the herpes simplex thymidine kinase (HSV-TK) gene. The HSV-TK gene is involved in the metabolism of antiviral ganciclovir and causes selective death of transfected cells when exposed to ganciclovir. First experience with TK lymphocytes was conducted in the context of hematological disease relapse and EBV^+^ PTLD as DLI after HLA-identical allogeneic HSCT. Survival of TK lymphocytes after infusion, GvL activity directly correlating with in vivo expansion of TK lymphocytes, and effective control of GvHD with ganciclovir were demonstrated [120,121]. Based on these results, TK lymphocytes were then tested in the haploidentical context with the aim of improving immune reconstitution and GvL as well as controlling GvHD. The phase I–II multicenter TK007 trial enrolled patients with hematological malignancies undergoing haploidentical HSCT with positive selection of CD34^+^ cells and no further GvHD prophylaxis after infusion. In the absence of valid immune reconstitution, 28 out of 50 enrolled patients received TK lymphocytes at escalating doses. Engraftment of TK lymphocytes was reported in 22 patients shortly after the first infusion. All-grade acute GvHD occurred in 10 patients, while 1 patient developed chronic extensive GvHD, both effectively controlled with ganciclovir. At 3 years, NRM and OS were 40% and 49%, respectively, in patients with acute leukemia in complete remission at HSCT [122]. Notably, TK lymphocytes have been proven to support also the long-term reconstitution of polyclonal unmanipulated lymphocytes by positive modulation of thymic functions [123]. A similar approach was carried out in the setting of haploidentical HSCT with donor T lymphocytes engineered with the inducible caspase 9 suicide gene (iC9-T cells). The iC9-T cells could be eliminated by administration of a chemical inducer of dimerization (AP1903). Compared to TK lymphocytes, this different mechanism allowed a more rapid inactivation of engineered cells and the possibility to receive antiviral drugs without T-cell damage. All 12 enrolled patients who underwent haploidentical HSCT and received iC9-T cells obtained robust immune reconstitution against viral and opportunistic infections. The administration of AP1903 in four patients with GvHD granted rapid and highly effective clearance of iC9-T cells from both peripheral blood and the central nervous system [124]. Long-term persistence of iC9-T as well as a positive immunological effect on polyclonal unmanipulated T lymphocytes were observed also with this approach, reflecting sustained protection from infectious complications [125]. The clinical trials in haploidentical transplantation setting are summarized in Table 5.

## 3. Conclusions

The cell therapy platforms associated with stem cell transplantation discussed here have been shown to be able to successfully prevent GvHD and, at the same time, allow a potent antileukemic effect (Figure 2). The proposed immunotherapies could represent the solution capable of reducing the incidence of post-transplant relapses. At the moment, among the cellular strategies illustrated in the present work, the most promising appear to be TCR alpha/beta depletion and transplantation with Treg/Tcon-adoptive immunotherapy. Both strategies are associated with the prevention of GvHD and a powerful antileukemic effect especially evident in the platform that includes the use of Tregs cells. However, the cellular manipulation present in these procedures requires adequate standardization between the various centers in order to guarantee wider use.

## Figures and Tables

**Figure 1 cells-13-00134-f001:**
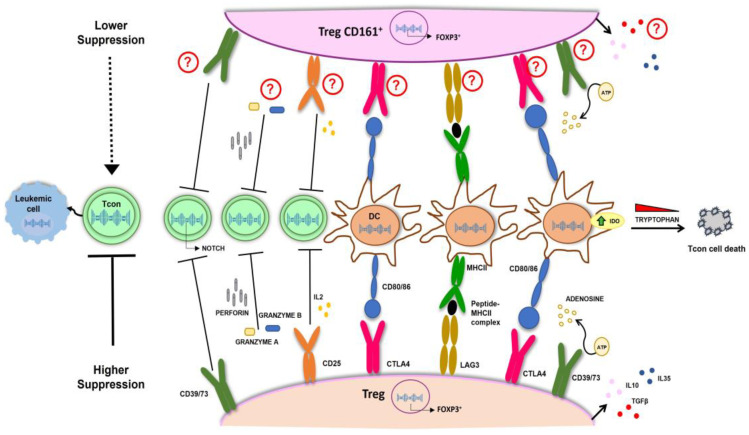
Main mechanisms of action of Tregs cells involve cytokine production, such as IL-10, IL-35, and TGF-beta; direct killing of Tcons by perforin/granzyme mechanism; IL-2 competition; Tregs/DC interaction with downregulation of CD80/CD86 on DCs via CTLA-4; interference with DC maturation through the LAG-3; production of IDO in DC through CTLA-4 and CD80/86 interaction which accelerates tryptophan degradation fundamental for Tcon survival; ATP cleavage by CD39/73; NOTCH1 downregulation in Tcons through CD39; ATP cleavage by CD39/73; NOTCH1 downregulation in Tcons through CD39. Although CD161^+^ Tregs showed a lower suppression activity on T cons compared to Tregs, the mechanisms involved are currently under investigation. Tregs, Regulatory T cells; Tcons, conventional T cells; DCs, Dendritic Cells; IDO, Indoleamine 2,3-dioxygenase.

**Figure 2 cells-13-00134-f002:**
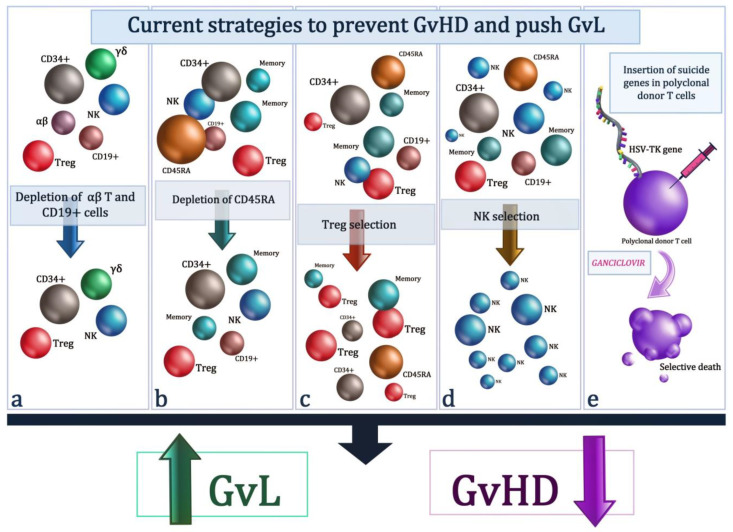
Current strategies to prevent GvHD and GvL include (**a**) the depletion of αβ T and CD19^+^ B cells from mobilized peripheral blood stem cells produces the generation of an allogeneic graft enriched for CD34^+^ stem cells and γδ T lymphocytes, preserving the NK compartment; (**b**) the depletion of I T cells (CD45RA^+^) is depicted; (**c**) purified Tregs are separated and infused together with CD34^+^ hematopoietic stem cells, as well as with conventional T cells; (**d**) NK cells are selected and infuI; (**e**) the suicide gene therapy using the herpes simplex thymidine kinase (HSV-TK) gene is represented. The HSV-TK gene is involved in the metabolism of antiviral ganciclovir and causes the selective death of transfected cells when exposed to ganciclovir.

**Table 1 cells-13-00134-t001:** Clinical trials and outcomes after haplo-HSCT with depletion of alloreactive αβ T cells and CD19^+^ B cells.

Author	Diagnosis/ Patient Number	Donor	Conditioning	GvHD Prophylaxis	Graft	Survival	CI of aGvHD	CI of cGvHD
Bertaina, A. et al. Blood 2014 [18]	Children with malignant and nonmalignant diseases (n = 23)	haplo	MAC 30%; NMA 70%	ATG (n = 23)	CD34^+^ cells/kg: 15.8 × 10^6^ (range: 10.4 × 10^6^ to 40 × 10^6^) TCR-αβ^+^CD3^+^ cells/kg: 4 × 10^4^ (range: 1 × 10^4^ to 9.5 × 10^4^)	2y DFS 91.1% 2y OS 91.1%	Grade I–II: 13% Grade III–IV: 0%	18 months: 0%
Gaziev, J. et al. Blood Adv. 2018 [20]	Children with nonmalignant diseases (n = 14)	haplo	MAC 100%	CSA^+^ steroids (n = 12); CSA + MMF (n = 2)	CD34^+^ cells/kg: 15.7 × 10^6^ (range: 8.1 × 10^6^ to 39.2 × 10^6^) TCR-αβ^+^CD3^+^ cells/kg: 4 × 10^4^ (range: 1 × 10^4^ to 10 × 10^4^)	5y DFS 69% 5y OS 84%	Grade II–III: 28%	extensive cGvHD: 21%
Laberko, A. et al. Blood 2019 [21]	Children with nonmalignant diseases (n = 98)	MUD, haplo	MAC 74%, NMA 26%,	ATG + CSA ± MTX/MMF (n = 96)	CD34^+^ cells/kg: 10.5 × 10^6^ (range: 6.3 × 10^6^ to 14.9 × 10^6^) in haplo TCR-αβ^+^CD3^+^ cells/kg: 1.4 × 10^4^ (range: 0.5 × 10^4^ to 13 × 10^4^) in haplo	5y OS 86% in MUD; 5y OS 87% in haplo	Grade II–IV 17% in MUD; Grade II–IV 22% in haplo	limited cGvHD 9% in MUD; chronic GvHD 13% in haplo
Locatelli, F. et al. Blood 2017 [22]	Children with malignant diseases (n = 80)	haplo	MAC 100%	ATG (n = 80)	CD34^+^ cells/kg: 13.9 × 10^6^ (range: 6 × 10^6^ to 40.4 × 10^6^) TCR-αβ^+^CD3^+^ cells/kg: 4.7 × 10^4^ (range: 0.2 × 10^4^ to 9.9 × 10^4^)	5y DFS 71% 5y OS 72%	Grade I–II 30%	limited cGvHD 5%
Prezioso, L. et al. Bone Marrow Transpl. 2019 [27]	Adult with malignant diseases (n = 59)	haplo	MAC 100%	ATG (n = 59)	CD34^+^ cells/kg: 11 × 10^6^ (range: 5 × 10^6^ to 19 × 10^6^) TCR-αβ^+^CD3^+^ cells/kg: 8.4 × 10^4^ (range: 0.4 × 10^4^ to 62 × 10^4^)	2y OS 51%	Grade II–IV: 17% Grade III–IV: 3%	limited cGvHD 3%
Kaynar, L. et al. Hematology 2017 [28]	Adult with malignant diseases (n = 34)	haplo	MAC 100%	ATG (n = 34)	CD34^+^ cells/kg: 12.7 × 10^6^ (range: 10.3 × 10^6^ to 16 × 10^6^) TCR-αβ^+^CD3^+^ cells/kg: 1.8 × 10^4^ (range: 0.7 × 10^4^ to 2.5 × 10^4^)	2y DFS 33%; 2y OS 36%	Grade I–IV: 30.3% Grade III–IV: 6.1%	cGvHD 6.1%

MAC: myeloablative conditioning regimen; NMA: non-myeloablative conditioning regimen; MUD: matched unrelated donor; ATG: anti-thymocyte globulin; CSA: cyclosporine; MMF: mycophenolate mofetile; MTX: methotrexate.

**Table 2 cells-13-00134-t002:** Clinical trials and outcomes after haplo-HSCT with CD45RA-depleted progenitor cell grafts.

Author	Diagnosis/ Patient Number	Donor	Conditioning	GvHD Prophylaxis	Graft	Survival	CI of aGvHD	CI of cGvHD
Naik, S. et al. Blood 2021 [47]	Children with hematologic malignancies (n = 72)	haplo	subMA 100%	MMF (n = 61) and/or sirolimus (n = 8)	Day 0: CD34^+^ cells/kg: 9.85 × 10^6^ (range: 1.96 × 10^6^ to 44.64 × 10^6^) day +1: CD45RA-depleted graft: CD34^+^ cells/kg: 5.82 × 10^6^ (range: 0.58 × 10^6^ to 39.43 × 10^6^) CD3^+^ T cells/kg: 60.1 × 10^6^ (range: 16.08 × 10^6^ to 528.43 × 10^6^) CD3^+^CD45RA^+^ cells/kg: median 0, range 0–0.2 × 10 ^6^ cells/kg). day +6: NK cells (median: 11.7 × 10 ^6^ cells/kg; range: 1.65–99.2)	3-y OS: 68.9% 3-y DFS: 62.2%	Grade II–IV: 36.1% Grade III–IV: 29.2%	3y: 20.8%
Sisinni, L. et al. Biol. Blood Marrow Transpl. 2018 [43]	Children with acute leukemias (n = 25)	haplo	subMA 100%	CSA (n = 3), CSA^+^ MTX (n = 1), MMF (n = 21)	CD34^+^ cells/kg: 6.29 × 10^6^ (range: 4.04 × 10^6^ to 18.1 × 10^6^) CD45RA^+^ cells/kg: 0.6 × 10^4^ (range: 0.2 × 10^4^ to 1 × 10^4^)	30 months OS: 58%	Grade II–IV: 39% Grade III–IV: 33%	30 months: 22%
Gasior Kabat, M. et al. Int. J. Hematol. 2021 [45]	Children with hematologic malignancies (n = 17) Severe aplastic anemia (n = 1)	haplo MRD	subMA 100%	MMF (n = 18)	CD34^+^ cells/kg: 6.5 × 10^6^ (range: 5 × 10^6^ to 11.2 × 10^6^) CD3^+^CD45RA^+^ cells/kg: 3.6 × 10^4^ (range: 0 to 23 × 10^4^) day +7: NK cells/kg: 12.6 × 10^6^ (range: 3.9 × 10^6^ to 100 × 10^6^)	2yOS: 87.2% 2y DFS: 67.3%	d+180: III–IV: 34.8%	1y: 23.1%

MA: myeloablative conditioning; MMF: mycophenolate mofetile; CSA: cyclosporine; MRD: matched related donor.

**Table 3 cells-13-00134-t003:** Clinical trials and outcomes after CD45RA-depleted donor lymphocyte infusions after haploidentical HCT.

Authors	Diagnosis/ Patient Number	Transplantation Platform	Cell Dose	Survival	CI of aGvHD	CI of cGvHD
Dunkaina, M. et al. BMT 2021 [49]	Children with hematologic malignancies (n = 143)	TCR αβ depletion MAC conditioning Haplo (n = 69) MUD (n = 6) MSD (n = 1)	Prophylactic—day 0: 25 × 10^3^ cell/kg, day 30, 60, 90, 120 50 × 10^3^ cell/kg Median number of DLI given = 4 (range: 1–5)	2y OS: 79% 2y DFS: 72%	Grade II–IV: 14.5% Grade III–IV: 8%	2y: 6%
Naik, S. et al. Blood 2021 [47]	Children with acute leukemia (n = 30)	TCR αβ depletion	Prophylactic two weeks following engraftment: DL1: 1 × 10^5^ cells/kg DL2: 1 × 10^6^ cells/kg DL3: 1 × 10^7^ cells/kg	1y OS: 86.3% 1y DFS:69.8%	Grade II–IV: 26.7% Grade III–IV: 13.3%	None
Castagna, L. et al. Transpl. Cell Ther. 2021 [50]	Adults with hematologic malignancies (n = 19)	Post-transplant cyclophosphamide; MAC/RIC	Prophylactic DLI of 3 infusions each 4–6 weeks apart First dose given at median of 55 days (range, 46–63) post HCT DL1 5 × 10^5^ cells/kg DL2 1 × 10^6^ cells/kg DL3 5 × 10^6^ cells/kg	1y OS: 79% 1y DFS: 75%	Grade I–IV 6%	1y: 15%

MAC: myeloablative conditioning; RIC: reduced-intensity conditioning; MSD: matched sibling donor; MUD: matched unrelated donor.

**Table 4 cells-13-00134-t004:** Clinical trials and outcomes after haplo-HSCT with NK cells.

Author	Diagnosis/ Patient Number	Donor	Conditioning	GvHD Prophylaxis	Cell Source and Dose	Survival	CI of aGvHD	CI of cGvHD
Choi, I. et al. Biol. Blood Marrow Transpl. 2014 [105]	Adults with hematological malignancies, mostly AML (n = 41)	haplo	RIC	CSA (n = 13); MTX + CSA (n = 28)	Donor-derived NK cells Infusion 2 and 3 weeks after transplant Escalating doses (median dose of 2.0 × 10^8^/kg)	31.5 months DFS: 31% OS: 35% (AML)	22%	24%
Ciurea, S.O. et al. Blood 2017 [114]	Adults with high-risk myeloid malignancies (n = 13)	haplo	RIC	CY (50 mg/kg per day on days 13 and 14) + tacrolimus from day 15 and for 6 months +MMF from day 15 and for 3 months	Membrane-bound IL-21 expanded donor NK cells Doses ranging from 1 × 10^5^/kg to 3 × 10^8^ kg)	1 year OS: 92% 1 year DFS: 85%	Grade I–II: 54% Grade III–IV: 0	0
Berrien-Elliot, M.M. et al. Sci. Transl. Med. 2022 [110]	Adult patients with high-risk AML (n = 15)	haplo	RIC	tacrolimus and MMF starting on day +5 (until days +180 and +35, respectively)	Donor-derived memory-like NK cells. Doses ranging from 0.5 × 10^6^ to 10 × 10^6^ cells/kg IL 15 agonist administered subcutaneously on day +7 and over 3 weeks.	1y OS: 29%	Grade I: 26% Grade II: 40%	All grade: 13%
Shapiro, R.M. et al. J. Clin. Investig. 2022 [119]	Relapsed myeloid malignancies after haplo HCT (n = 6)	haplo	RIC	ATG + tacrolimus + MTX (n = 21), ATG + tacrolimus (n = 5), ATG + MTX (n = 2), ATG (n = 5)	Donor-derived memory like NK cells Dose ranging from 5 to 10 × 10^6^ cells/kg + IL2 (7 doses)		All grade: 0	All grade: 0

CSA: cyclosporine; MTX: methotrexate; AML: acute myeloid leukemia; RIC: reduced-intensity conditioning; CY: cyclophosphamide; MMF: mycophenolate mofetile; ATG: anti-thymocyte globulin.

**Table 5 cells-13-00134-t005:** Clinical trials and outcomes after haplo-HSCT with suicide gene therapy.

Author	Diagnosis/Patient Number	Donor	Conditioning	GvHD Prophylaxis	Graft Composition	Survival	CI of aGvHD	CI of cGvHD
Ciceri, F. et al. Lancet Oncol. 2009 [122]	Adult with malignant diseases (n = 50) 22 received TK cells	haplo	MAC	ATG (n = 45)	CD34^+^ cells/kg: 11.6 × 10^6^ (range: 4.6 × 10^6^ to 16.8 × 10^6^) CD3^+^ cells/kg: 1 × 10^4^ (range: 0.26 × 10^4^ to 10 × 10^4^)	3y OS 49%	Grade I–IV 45%	Extensive cGvHD 4%

MAC: myeloablative conditioning regimen; ATG: anti-thymocyte globulin; TK: lymphocytes expressing herpes simplex thymidine kinase suicide gene.

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
