# Peer review of "Cellular Strategies for Separating GvHD from GvL in Haploidentical Transplantation"

_cells, 2024, doi:10.3390/cells13020134_

Round 1
Reviewer 1 Report
Comments and Suggestions for Authors
This is a comprehensive review of cellular platforms to use in conjunction with haploidentical transplantation to separate GVHD from GVL.
Comments:
This is also important in other types of allogeneic transplantation, so it is uncertain why the focus on haploidentical transplants . It would be important to describe this in the discussion and to discuss how post-transplant cyclophosphamide alters the immune milieu providing opportunity to utilized the cellular therapies described here.
The text is quite dense and could benefit from additional paragraph breaks with transitional sentences.
The tables are quite complete, but the formatting makes comparisons difficult.
The conclusion would benefit from some future directions. How does one determine the best approach for GVHD/GVL separation and how will the field define this and arrive at this in the future?
Minor: Define Tcons when first appears.
Comments on the Quality of English LanguageThere are several places where plurals are incorrect, etc. , so some editing is needed.
Author Response
First of all we would like to thank the reviewer for her/his valuable advice. As an initial choice we decided to focus on transplant platforms where there is cellular manipulation. Therefore, transplantation from a haploidentical donor is the one with the greatest possibilities described in the literature. For the same reason we have not discussed the PTCY platform, in which GvHD prophylaxis is entrusted to pharmacological therapy with cyclophosphamide and not to specific cell manipulation techniques which are the subject of this review.
As suggested by the reviewer, we have included transitional sentences in order to make reading the manuscript easier. All changes are highlighted in yellow.
We modified the tables making them more homogeneous in the representation of the data. The original Table 2 has been divided into Table 2A and Table 2B. All changes made in the text are highlighted in yellow.
As requested, we have included some final thoughts on future prospects and how GvHD/GvL separation can best be achieved.
Tcons (i.e. conventional T cells) were defined as suggested
Reviewer 2 Report
Comments and Suggestions for Authors
This is a well-written review article to summarize the current literature and strategies to prevent GvHD and maximally amplify the GvL effect in the setting of haploidentical transplantation.
To separate GvL and GvHD, the review illustrated the use of TCR alpha/beta depletion, CD45 RA depletion, T regulatory cells enrichment, NK cell-based immunotherapies, and suicide gene therapies to prevent GvHD and maximally amplify the GvL effect in the setting of haploidentical transplantation. I would like to accept this manuscript with appropriate revision.
1. B cells may also play an important role in GVL. It is well described that both major histocompatibility complex and minor histocompatibility antigens can elicit B-cell antibody responses.
2. The GVL effect post-HCT recipients with moderate to severe chronic GVHD (cGVHD) were significantly less likely to relapse. The most recent comprehensive analysis to evaluate the relative roles of both aGVHD and cGVHD on the GVL effect. Please comment on how to analyze the phase of acute versus chronic GVHD.
Author Response
First of all we would like to thank the reviewer for her/his valuable advice. The presence of circulating donor specific alloantibodies may sustain primary graft failure in HLA-mismatched allografts (Bettinotti et al., Hematology Am Soc Hematol Educ Program. (2017) 1:645–50). It is possible that alloantibodies may also play a role in disease remission. Studies have shown a highly significant association between H-Y antibodies and decreased relapse in male patients with female donors. However, this effect is also directly related to increased rates of chronic GVHD (Rozmus et al., Front Ped 2022). In any case, the platforms covered by this review almost always include B cell depletion (TCR alpha/beta and Treg selection). Consequently, we believe that the discussion on the role of B cells between GvHD and GvL is not part of the present review.
The incidence of GVHD following allogeneic HSCT is reported as 30–50% for acute and 30–70% for chronic disease (Zeiser et al NEJM 377, 2167-79, 2017 and Zeiser et al. 377, 2565-79, 2017). Furthermore, the development of cGVHD was associated with a higher risk of NRM, lower risk of relapse and longer OS, in patients undergoing matched allogeneic Hematopoietic Cell Transplantation (Bhatt et al., TCT 2022). In the present review we analyzed the different cell separation strategies used to separate GvHD and GvL in the setting of haploidentical transplantation. For each illustrated strategy we have reported the CI of aGvHD and cGvHD in the different tables. The different strategies are not comparable, as the cells involved are different for each platform. In Treg transplantation (Di Ianni et al., Blood 2011) the incidence of cGvHD is close to zero. The data was confirmed by the recent paper by Pierini et al. (Blood Adv, 2022). This platform simultaneously presents a very powerful GvL effect. In other words, when the platform used includes a cell separation method, the mechanisms involved in the determinism of GvHD/GvL may not be completely superimposable with those of non-manipulated transplantation.